# A CMOS Bandgap-Based VCSEL Driver for Temperature-Robust Optical Applications

**Juntong Li and Sung-Min Park ***

Division of Electronic & Semiconductor Engineering, Ewha Womans University, Seoul 03760, Republic of Korea; lijuntong@ewhain.net

* Correspondence: smpark@ewha.ac.kr

**Abstract**

This paper presents a temperature-robust current-mode vertical-cavity surface-emitting laser (VCSEL) driver (or CMVD) fabricated in a standard 180 nm CMOS process. While prior art relies on conventional current-mirror circuits for bias generation, the proposed CMVD integrates a bandgap-based biasing architecture to achieve high thermal stability and process insensitivity. The bandgap core yields a temperature-compensated reference voltage and is then converted into both stable bias and modulation currents through a cascode current-mirror and switching logic. Post-layout simulations of the proposed CMVD show that the reference voltage variation remains within ±2%, and the bias current deviation is under 10% across full PVT conditions. Furthermore, the output current variation is limited to 7.4%, even under the worst-case corners (SS, 125 °C), demonstrating the reliability of the proposed architecture. The implemented chip occupies a compact core area of 0.0623 mm$^2$ and consumes an average power of 18 mW from a single 3.3 V supply, suggesting that the bandgap-stabilized CMVD is a promising candidate for compact, power-sensitive optical systems requiring reliable and temperature-stable performance.

**Keywords:** bandgap; current-mode; driver; LiDAR; VCSEL; temperature

## 1. Introduction

Over the past decades, short-range optical transmission systems have become increasingly vital in consumer electronics, biomedical sensors, and environmental monitoring. These applications often demand compact, low-power, and thermally reliable optical transmitters. Among various laser sources, vertical-cavity surface-emitting laser (VCSEL) diodes have gained significant attention due to their low threshold current, wafer-level testability, and compatibility with CMOS integration.

However, one major challenge in CMOS-based VCSEL driver designs is the impact of process, voltage, and temperature (PVT) variations, which can degrade current accuracy and modulation linearity. Conventional voltage-mode or current-mode drivers are often insufficient to guarantee stable output performance across PVT corners because conventional implementations often require external trimming or complex feedback loops to maintain performance over PVT variations.

It is well known that bandgap-based bias circuits have been widely employed to provide temperature-compensated reference voltages for analog and mixed-signal systems. Hence, to alleviate the aforementioned limitations of conventional drivers, the proposed current-mode VCSEL driver (CMVD) integrates a compact current-mode VCSEL driver

that integrates a Brokaw cell-based bandgap core with digital current-steering logic, thereby resulting in enhanced stability across process and temperature variations [1,2].

Figure 1 illustrates an architecture of a short-range LiDAR sensor system that is designed to be installed on ceilings of constrained indoor environments, such as single-room shelters, vinyl housing units, or semi-basement dwellings where children affected by housing poverty reside [3]. It transmits near-infrared light pulses through a laser diode toward occupants (i.e., mostly a child under 18 years old) and detects the reflected signals by using an avalanche photodiode (APD) to determine occupancy, movement, or posture changes in real time. Specifically, the system measures the time-of-flight (ToF)—the delay between the transmitted and reflected pulses—to compute the vertical distance to the subject (Figure 1b). After emission, the laser pulse reflects off the target and is received by the APD. The returned signal is amplified and processed through a transimpedance amplifier (TIA), a single-to-differential (S2D) converter, a post-amplifier (PA), and an output buffer (OB), before being digitized by a time-to-digital converter (TDC) that records the round-trip delay. By analyzing these delays, the system extracts posture and movement patterns. This method, with a fixed downward emission angle, enables passive and privacy-conscious tracking of targets in vertically constrained indoor environments without requiring mechanical scanning [4].

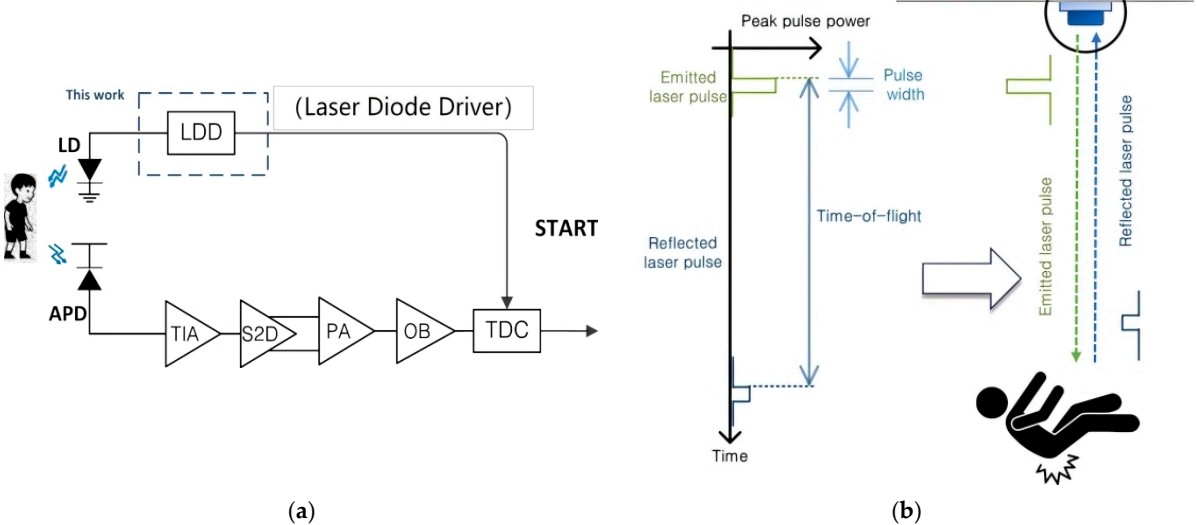

(a)　　　　　　　　　　　　　　　　　　　　　　　　　　　　(b)

**Figure 1.** (**a**) Block diagram of a typical LiDAR sensor and (**b**) time-of-flight (ToF)-based distance measurement.

Among the various laser sources for LiDAR sensor systems, vertical-cavity surface-emitting laser (VCSEL) diodes have gained widespread adoption in short-range, power-sensitive situations due to their vertical light emission, simplified packaging, and compatibility with wafer-level testing. Unlike edge-emitting lasers (EELs), which require complex alignment and higher operating voltages, VCSEL diodes offer a favorable tradeoff between cost, manufacturability, and system integration. According to Pan et al. [5], VCSEL diodes outperform other types of lasers in terms of manufacturing simplicity, reliability, and eye safety, making them especially well-suited for indoor applications where safety and affordability are critical.

Despite these advantages, VCSEL diodes exhibit a limitation in their optical range compared to fiber or edge-emitting counterparts. Yet, for the specific application of room-scale child behavior monitoring, the required detection range is well within the operating capability of VCSEL diodes. In addition, their low threshold current and symmetric emission characteristics allow for efficient beam shaping without expensive optics. From a circuit design perspective, the moderate forward voltage requirement (typically ~1.5 V)

makes VCSEL diodes compatible with standard CMOS driver implementations, provided that careful attention is paid to the voltage-headroom and the ESD protection constraints. Furthermore, a common-cathode VCSEL topology is employed in this work to simplify supply management and ensure its compatibility with single-rail CMOS biasing schemes.

## 2. Conventional Voltage-Mode VCSEL Driver

Figure 2 depicts a typical implementation of voltage-mode VCSEL drivers with common-cathode configurations, where the modulation signal is processed through a chain of stages beginning with a bias and equalization block (IB + EQ), followed by a pre-amplifier and a main driver that directly drives the VCSEL diode. These voltage-mode drivers have been favored for their simplicity and compatibility with differential signaling standards such as current-mode-logic (CML) or low-voltage differential signaling (LVDS). However, the equalization stage is often implemented with passive resistive–capacitive (RC) networks. Therefore, they are highly sensitive to process variation and temperature fluctuations. Besides, the use of voltage-mode switching at the final driver stage leads to a significant signal swing, thereby leading to an enlarged power dissipation and slow edge transitions under certain capacitive loads. Provided that the biasing and equalization logic is merged into a minimal bias unit (IB) only, the overall performance suffers from the inherent difficulty of precisely regulating the output current—especially in short-range, low-power applications where modulation linearity and stability are critical. Moreover, this architecture mandates careful layout to mitigate the parasitic capacitance at the driver-to-VCSEL interface and exhibits its limited adaptability to temperature-induced current drift. This becomes particularly problematic in indoor environments with poor thermal regulation, such as those found in housing-poverty scenarios.

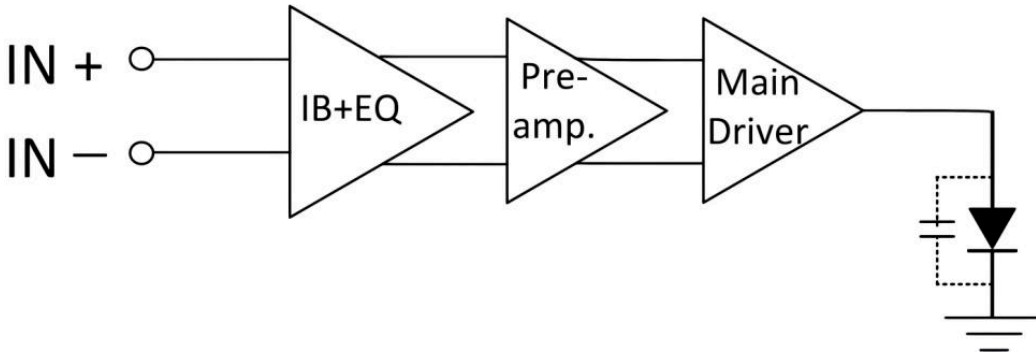

**Figure 2.** Block diagram of a conventional voltage-mode VCSEL driver [6,7].

In a typical sensing-grade single-emitter VCSEL, the threshold current is on the order of 0.75–3 mA (up to ~5 mA depending on wavelength and structure), and the operating/peak modulation current for short-range ToF applications commonly falls in the range of ~3–15 mA$_{pp}$. Accordingly, our driver targets a ~3–4 mA bias and 8–10 mA$_{pp}$ peak modulation to meet the optical power while maintaining eye-safety margins.

## 3. Conventional Current-Mode VCSEL Driver

Figure 3 illustrates a conventional current-mode VCSEL driver architecture, where the core of this topology relies on a current-mirror circuit so that a reference bias current ($I_{IN}$ through M1) can be replicated toward the output branch to drive the VCSEL diode. The modulation signal is applied through a control voltage ($V_{CTRL}$), which toggles a cascode-configured NMOS switch (M2) to enable (or disable) the current flow to the VCSEL diode. The input current ($I_{IN}$) is typically sourced from an upstream biasing block and is mirrored through a pair of matched PMOS transistors (M7, M8). The presence of

stacked PMOS transistors (M8, M9) enhances the output impedance and the modulation linearity. This cascode configuration mitigates drain-induced variations at the output stage. Such improvements in current-mirror accuracy and linearity ensure consistent optical modulation of the VCSEL didoes, thereby reducing pulse distortion and timing jitter, which is essential for reliable ToF measurement accuracy. Two inverter chain (M3–M6) is inserted to turn on the cascode M9 only with low state of $V_{CTRL}$.

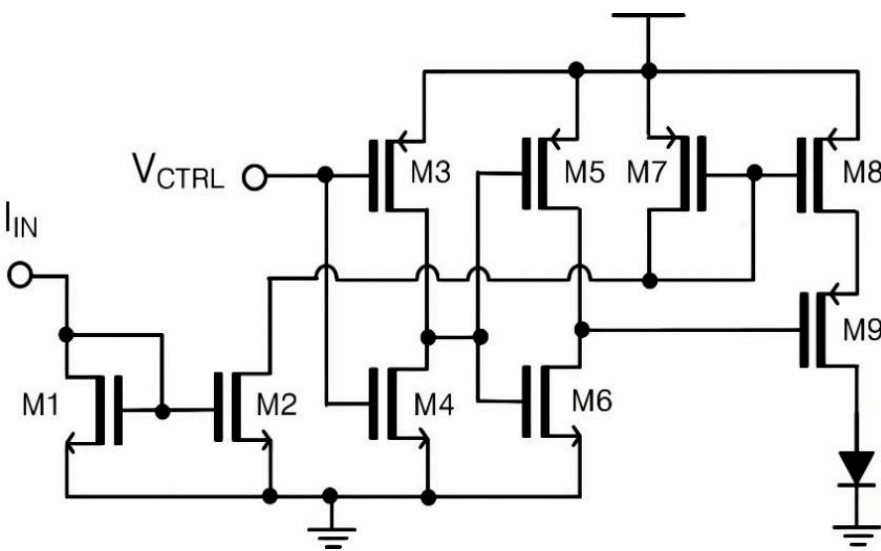

**Figure 3.** Schematic diagram of a conventional current-mode VCSEL driver using a common-cathode configuration [8].

Despite their improved current control capability compared to voltage-mode drivers, current-mode drivers suffer from several practical limitations. First, due to the absence of an active feedback loop, the output current remains sensitive to PVT variations, unless carefully compensated. Second, the headroom required by the stacked transistors (M8, M9) limits its operation at a low supply voltage—posing challenges for low-power system design. Third, mirror accuracy (M1, M2 and M7, M8) is susceptible to mismatches in layout or device aging, which could cause considerable current drift over time.

To alleviate these drawbacks, the proposed architecture in this work introduces a bandgap-stabilized current-mode VCSEL driver that enhances thermal robustness and current accuracy by combining regulated reference generation with digitally programmable modulation switches. The details of the proposed structure are presented in the next section.

## 4. Proposed Bandgap-Stabilized Current-Mode VCSEL Driver

As previously mentioned, this work proposes a thermally robust current-mode driver architecture integrated with a bandgap-based bias generation circuit to overcome the limitations inherent in both voltage-mode and conventional current-mode VCSEL drivers. Figure 4 shows the block diagram of the proposed transmitter named 'a bandgap-stabilized current-mode VCSEL driver (CMVD)' that leverages a highly stable reference current derived from an on-chip bandgap core to ensure minimal drift over severe PVT variations. This CMVD can inherently regulate the output current through mirrored reference sources with digitally switchable branches. By relying on thermally stabilized bias currents, the proposed architecture can achieve higher modulation fidelity, improved temperature robustness, and reduced overall power consumption.

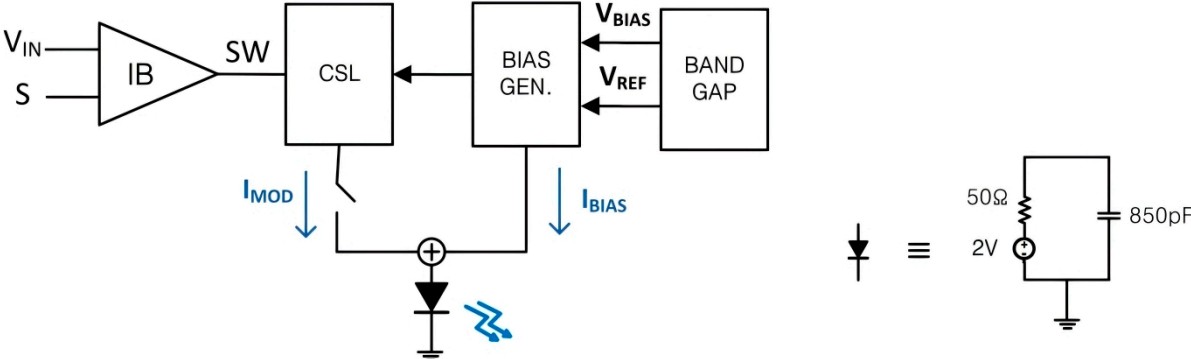

**Figure 4.** Block diagram of the proposed CMVD and the equivalent circuit model of a VCSEL diode.

The CMVD consists of a bandgap reference circuit, a bias generator, and a modulation signal from the IB and selectively routes the modulation current ($I_{MOD}$) through the VCSEL diode. Here, 'S' denotes the digital modulation control signal. In parallel, the bias current ($I_{BIAS}$) is continuously supplied to maintain the VCSEL's threshold-level emission. This topology enables binary-weighted (or pulse-driven) modulation by digitally switching current branches without an analog control overhead.

To emulate practical optical load conditions and evaluate signal integrity, an external termination network—comprising a 50 Ω resistor and an 850 fF capacitor—is implemented on the PCB and connected to the driver output. This configuration facilitates accurate measurements of transient behavior, including rise and fall times, and power consumption. A more detailed description of each circuit block follows below, highlighting transistor-level implementation, layout methodology, and robustness against PVT variations [9–13].

In a conventional driver (shown in Figure 3), both mismatches and PVT drifts contribute to a large $\sigma_I$, thus leading to degraded ToF precision. However, the proposed bandgap-based driver (shown in Figure 4) suppresses PVT-induced drift, thereby reducing $\sigma_I$ and enhancing the overall ToF accuracy. To quantify this relation, the ToF timing jitter can be approximately given by

$$\sigma_I \approx \frac{\sigma_t}{dI/dt},$$

where $\sigma_t$ is the timing uncertainty, $\sigma_I$ is the fluctuation of the driving current, and $dI/dt$ is the slew rate of the modulation edge.

**(1)  Bandgap Reference Circuit**

The bandgap reference block (illustrated in Figure 5) provides two temperature-compensated outputs, i.e., a regulated reference voltage ($V_{REF}$) and a bias voltage ($V_{BIAS}$). Here, $V_{BIAS}$ not only serves as an output, but is also internally utilized to bias the subsequent circuits, thus ensuring stable operations across PVT variations. This architecture shares a classic Brokaw cell configuration that incorporates two bipolar transistors (T1, T2) with differing emitter areas. These transistors produce a combination of a complementary-to-absolute-temperature (CTAT) voltage and a proportional-to-absolute-temperature (PTAT) voltage. The voltage drop across resistor R1, representing the PTAT component, is defined by

$$V_{PTAT} = V_{BE1} - V_{BE2} = \frac{KT}{q}\ln\left(\frac{A_2}{A_1}\right), \tag{1}$$

where $A_2/A_1$ is the emitter area ratio of the two BJTs.

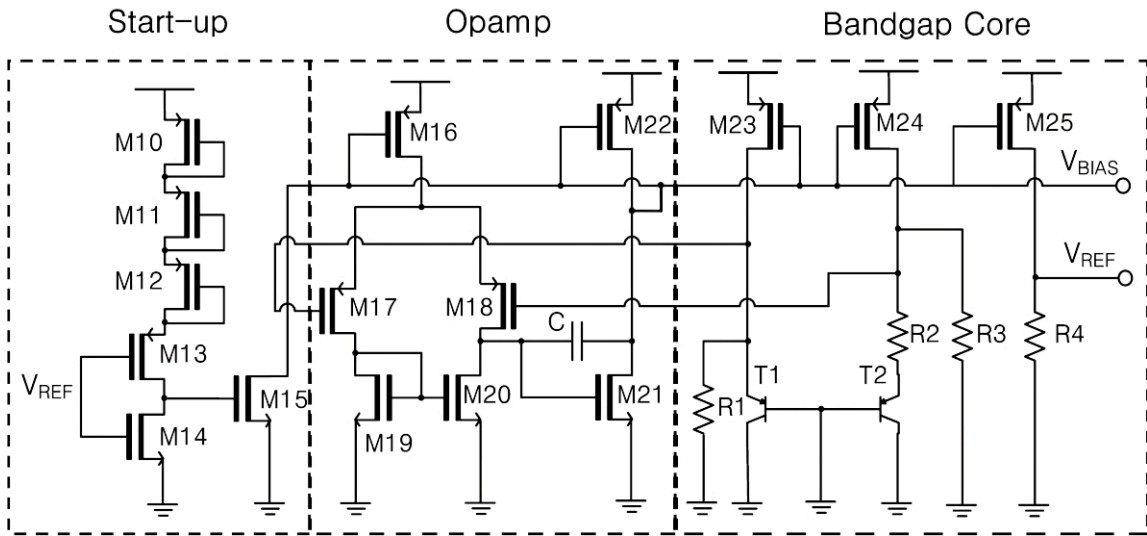

**Figure 5.** Schematic diagram of the bandgap reference circuit.

The PTAT voltage is scaled through a resistor network (R2–R4) and summed with the base-emitter voltage to yield a composite voltage with minimal temperature dependency, i.e.,

$$V_{REF} \approx V_{BE} + m \cdot V_{PTAT} , \qquad (2)$$

where $m = k_{mirror}$ (R4/R2), with $k_{mirror}$ being the effective current-mirror gain from the PTAT branch to the output branch, determined by the sizing of M23–M25.

This reference voltage ($V_{REF}$) is then distributed to downstream blocks after being buffered and mirrored through a chain of PMOS current mirrors (M22–M25). The current mirrors are responsible for delivering both $V_{REF}$ and $V_{BIAS}$ to the bias generator.

To ensure reliable operation and reduce mismatch, the MOSFETs (M16–M25) in the reference circuit were designed with a sufficient channel length (i.e., $L \geq 1$ μm), thus preventing short-channel effects and minimizing offset voltages. The width (W) of each transistor was carefully selected based on the targeted bias current in each branch, thereby ensuring saturation operations and good matching [14–16]. Table 1 presents the device dimensions and parameter values of the bandgap reference circuit.

**Table 1.** Device dimensions and parameter values of the bandgap reference circuit.

| M10–M12 | 10 μm/1 μm | M13, M14 | 3 μm/1 μm | M15 | 15 μm/1 μm |
|---|---|---|---|---|---|
| M16 | 15 μm/1 μm | M17–M18 | 30 μm/5 μm | M19–M20 | 2 μm/1 μm |
| M21 | 7 μm/2 μm | M22 | 15 μm/1 μm | M23–M25 | 1.5 μm/1 μm |
| C | 17 fF | T2:T1 | 8:1 | R1 | 126.8 kΩ |
| R2 | 5 kΩ | R3 | 126.8 kΩ | R4 | 75 kΩ |
| Cut-off Region | M10–M15 | Saturation Region | M16–M25 | Active Region | T1–T2 |

**(2) Bias Current Generator and Current Steering Logic**

Figure 6 depicts a schematic diagram of the bias combined with the CSL circuit, where the bias current generation and the modulation control are merged into a unified structure. Especially at its core lies a two-stage operational amplifier (OP-AMP) consisting of transistors M26 to M32 and a compensation capacitor (C) that helps stabilize the feedback loop. The OP-AMP receives the temperature-compensated reference voltage $V_{REF}$ supplied

from the preceding bandgap reference block as a positive input (at the gate of M29) and the feedback voltage ($V_{FB}$) as a negative input (at the gate of M28).

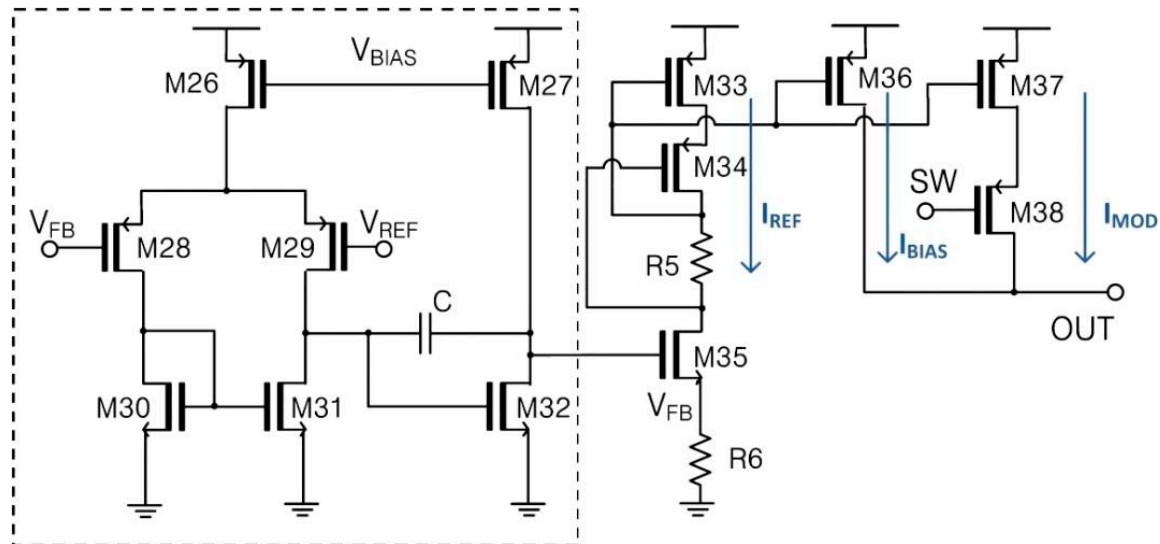

**Figure 6.** Schematic diagram of the bias and CSL circuit.

Then, the output of this OP-AMP regulates the transistor M35, thereby establishing the reference current ($I_{REF}$) through the resistor R6. The reference current is determined by

$$I_{REF} = \frac{V_{REF}}{R6} ,\tag{3}$$

where $V_{REF}$ is the temperature-compensated reference voltage provided by the bandgap core and R6 is a precision resistor connected in the OP-AMP feedback loop.

The stability of the bias regulation loop was verified using the loop gain simulation. The OP-AMP achieves a DC gain of approximately 80 dB and a phase margin of over 60°, hence ensuring reliable operation of the feedback.

This current ($I_{REF}$) is then mirrored via a cascode-configured PMOS mirror (M33, M34). The mirror branch through M36 delivers a steady bias current ($I_{BIAS}$) to the VCSEL diode, thus supporting its continuous baseline emission.

In parallel, a second mirror path branches through M37 and is gated by the modulation control signal SW via M38. When SW is active (i.e., logic high), this path is enabled, and the modulation current ($I_{MOD}$) is added to the final output. Thus, the total current driving the VCSEL diode becomes $I_{BIAS}$ plus $I_{MOD}$, enabling binary-weighted (or pulse-driven) modulation without analog control overhead.

Despite its simplicity, this switch-based modulation scheme effectively steers the desired currents in response to the digital control signals, and therefore enables low-complex yet precise optical modulations suitable for short-range LiDAR applications with low-power and high-speed requirements. Table 2 summarizes the device parameters of the bias and CSL circuit along with the performance characteristics of the op-amp.

**(3) Input Buffer**

Figure 7 shows a schematic diagram of the input buffer (IB) that is realized as a CMOS two-input AND gate (M39–M42) followed by an inverter (M43, M44). Here, two different inputs, i.e., the input signal ($V_{IN}$) and the modulation control signal (S), are jointly evaluated to generate the switching control signal (SW) at the output. Only when both inputs are high does SW go high, thus enabling the modulation path. This configuration

ensures that the modulation occurs only under the valid control signal, thereby preventing unintended current injection into the VCSEL diode.

**Table 2.** Device parameters and OP-AMP characteristics of the bias and CSL circuit.

| Item | Value | Unit | Item | Value | Unit |
|---|---|---|---|---|---|
| M26–M27 | 15 μm/1 μm | W/L | M28–M29 | 30 μm/5 μm | W/L |
| M30–M31 | 2 μm/1 μm | W/L | M32 | 7 μm/2 μm | W/L |
| M33–M34 | 15 μm/300 nm | W/L | M35 | 24 μm/1 μm | W/L |
| M36 | 600 μm/300 nm | W/L | M37 | 1500/300 nm | W/L |
| M38 | 2000 μm/300 nm | W/L | Capacitor (C) | 390 | fF |
| Resistor R5 | 8.1 | kΩ | Resistor R6 | 12.2 | kΩ |
| Linear Region | M26, M38 | — | Saturation Region | M27–M37 | — |
| DC Gain | ~80 | dB | UGB | ~400 | MHz |
| Phase Margin | >60 | degrees | PSRR@1 kHz | ~−35 | dB |
| Supply Voltage | 3.3 | V | Power Dissipation | 632.9 | μW |

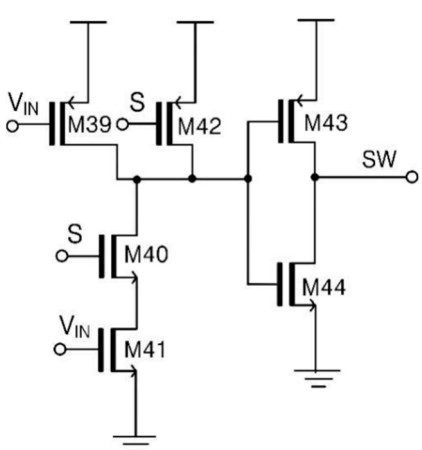

**Figure 7.** Schematic diagram of the IB circuit.

## 5. Layout and Simulation Results

Figure 8 depicts the layout of the proposed bandgap-based CMVD circuit, where the fabricated chip occupies a core area of 0.0623 mm². Post-layout simulations were conducted by using the model parameters of the TSMC 0.18 μm CMOS process, revealing that the CMVD consumes an average power of 18 mW from a single 3.3 V supply.

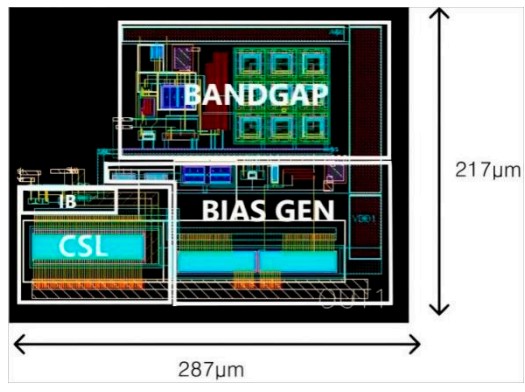

**Figure 8.** Chip layout of the proposed bandgap-stabilized CMVD.

To evaluate the power supply rejection capability of the bandgap reference, AC analysis was performed with sinusoidal perturbations injected at the supply node. As shown in Figure 9, the proposed CMVD circuit achieves more than 40 dB PSRR in the low-frequency region (i.e., below approximately 10 kHz), indicating strong immunity to supply noise at low frequencies.

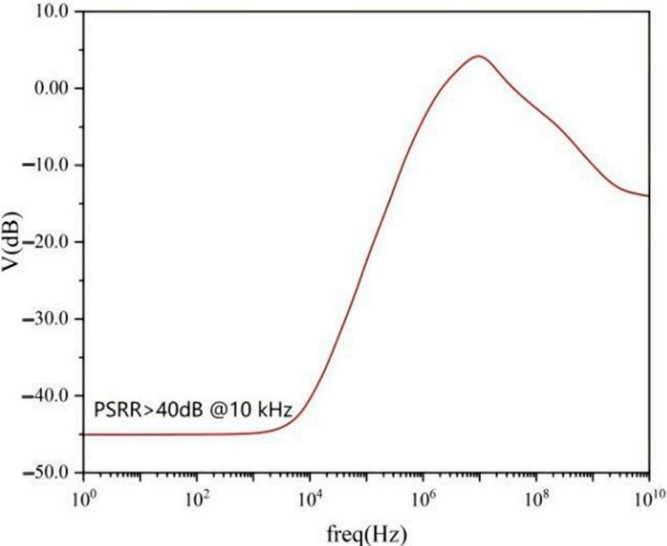

**Figure 9.** PSRR simulation results of the bandgap reference ($V_{REF}$).

Then, an open-loop AC simulation was performed to ensure the loop stability of the bandgap reference. As shown in Figure 10, the loop gain achieves a DC gain of approximately 65 dB with a unity-gain bandwidth of 100 MHz. The corresponding phase margin is around 85°, confirming the robust frequency-domain stability across the operational range. Pole–zero inspection of the return ratio indicates that the dominant pole at the op-amp output together with the Miller-path ESR-induced LHP zero cancels the mirror-gate pole, which provides effective first-order compensation. Furthermore, the second pole at the bandgap core node, assisted by a high-frequency feedforward LHP zero, establishes the second-order compensation, thereby ensuring robust loop stability across the operational range.

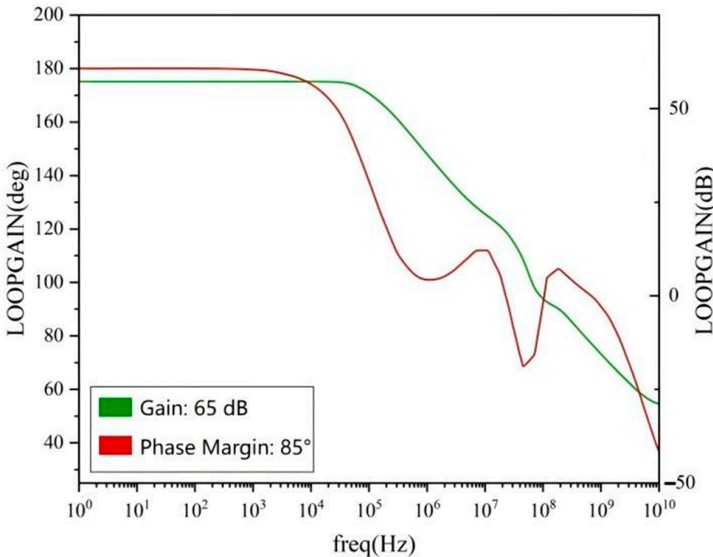

**Figure 10.** Loop gain and phase response of the bandgap reference feedback system.

Also, a DC temperature sweep from −55 °C to 125 °C was performed to evaluate the temperature stability of the bandgap output voltage. As shown in Figure 11, the reference voltage shows a clear curvature with a turning point near 50 °C, indicating the effective first- and partial second-order temperature compensation. The output varies by approximately 23.8% across the full temperature range, which corresponds to a temperature coefficient (TC) of about 1320 ppm/°C.

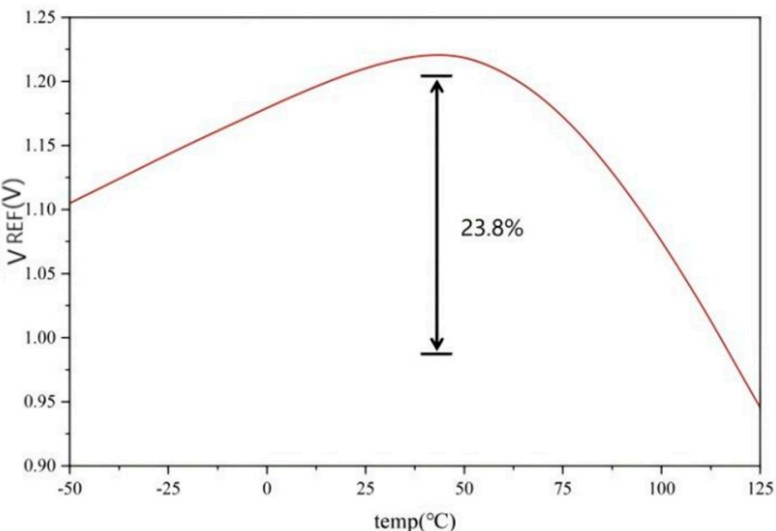

**Figure 11.** DC temperature sweep of the bandgap output voltage.

In addition, a DC temperature sweep from −55 °C to 125 °C was performed to verify the temperature stability of the CMVD bias current. As shown in Figure 12, the CMVD bias current exhibits a mild curvature with a peak near 80 °C, which indicates good thermal robustness. Since the overall variation remains within a stable range, it is suitable for reliable operations across the temperature extremes.

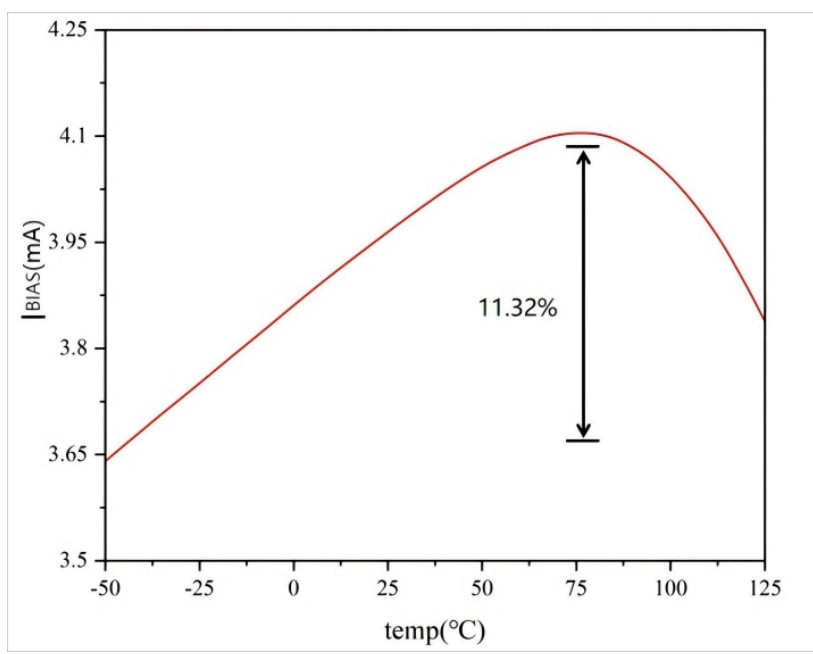

**Figure 12.** DC temperature sweep of the CMVD bias current.

Furthermore, a transient simulation was conducted to verify the power-up reliability with a 1 µs supply ramp and 2 ns delay across the temperature sweep. As shown in

Figure 13, the bandgap reference consistently starts up correctly under all the conditions, demonstrating robust cold- and hot-start behaviors during the fast power-up.

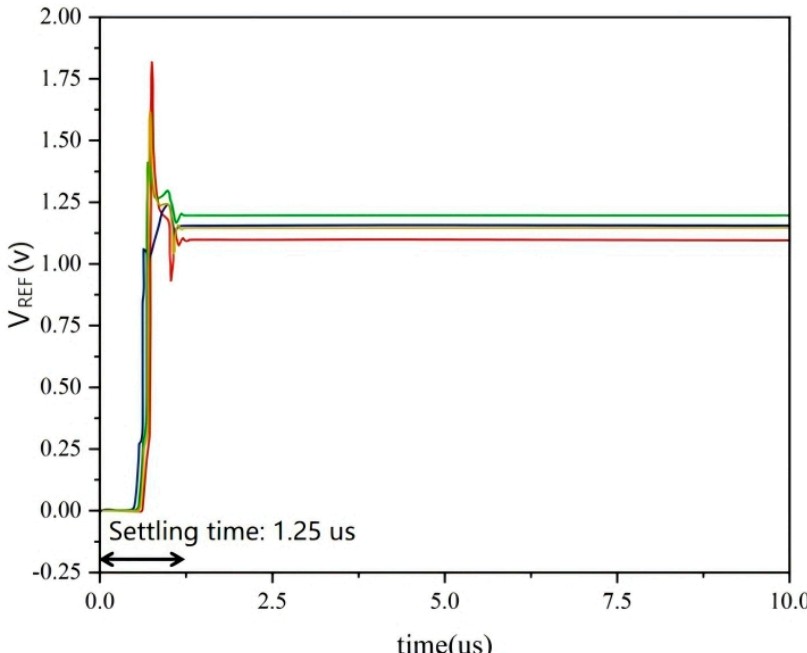

**Figure 13.** Transient startup response of the bandgap reference under fast power-up (1 μs ramp, 2 ns delay) across temperatures from −55 °C to 125 °C.

As observed, the reference voltage ($V_{REF}$) exhibits good temperature stability and process insensitivity in the post-layout simulation results. In the FF (fast-fast) corner case (shown in Figure 14a), taking the 25 °C value as nominal ($V_{REF,25\,°C} \approx 1.17$ V), $V_{REF}$ changes by approximately −1.7% at −55 °C and +1.7% at 125 °C. In the SS (slow-slow) corner case (shown in Figure 14c), the variation remains extremely narrow, fluctuating around 1.16 V, from ~1.14 V (−1.7%) to ~1.18 V (+1.7%).

These results demonstrate that the implemented bandgap core provides robust voltage stability under wide temperature and process variations. Specifically, the temperature-induced $V_{REF}$ drift stays within ±2% when the supply voltage is fixed at 3.3 V, thereby validating the suitability of this reference for precision biasing in analog front-end applications.

Moreover, the total $V_{REF}$ variation reaches approximately 5.8% when considering both temperature and supply voltage changes, which still demonstrates adequate robustness for low-voltage analog circuits.

Figure 15 illustrates the simulated bias current behavior of the proposed bandgap-stabilized CMVD over a wide range of PVT conditions. The *x*-axis denotes the swept supply voltage from 2.75 V to 4.0 V, while the *y*-axis indicates the resulting bias current levels measured at ambient temperatures ranging from −55 °C to 125 °C.

The simulation results reveal that the bias current exhibits moderate temperature dependence, varying with the considerable process corners. In the fast-fast (FF) corner case (Figure 15a), the output current shifts by −8.86% at −55 °C and +8.39% at 125 °C, relative to the nominal 4.29 mA observed at 25 °C. Under the slow-slow (SS) corner condition (Figure 15c), the bias current starts at 3.47 mA at room temperature and exhibits a fluctuation between −8.93% and +9.8% across the evaluated temperature range.

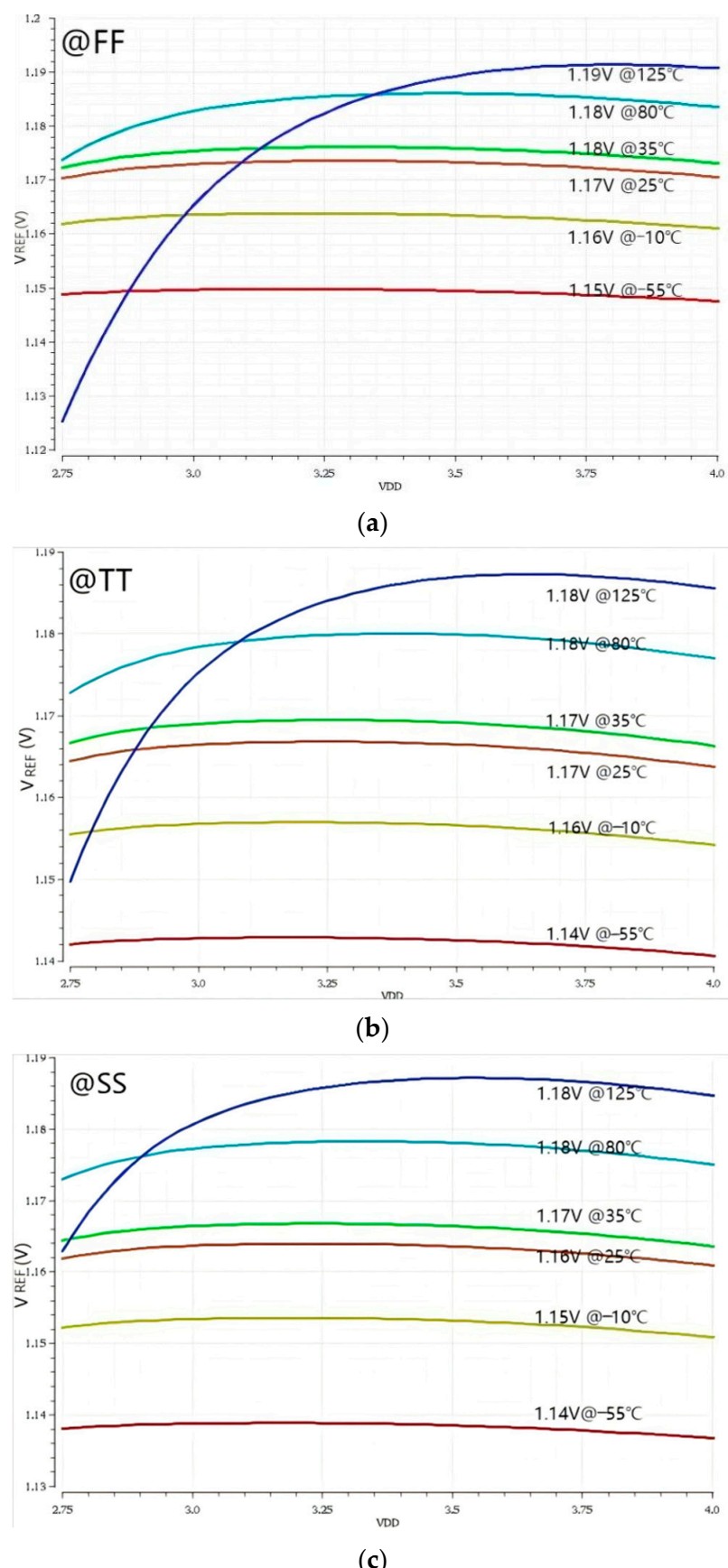

**Figure 14.** Simulated PVT variations of the reference voltage (V$_{REF}$) in the bandgap circuit for (**a**) FF, (**b**) TT, and (**c**) SS corners, respectively.

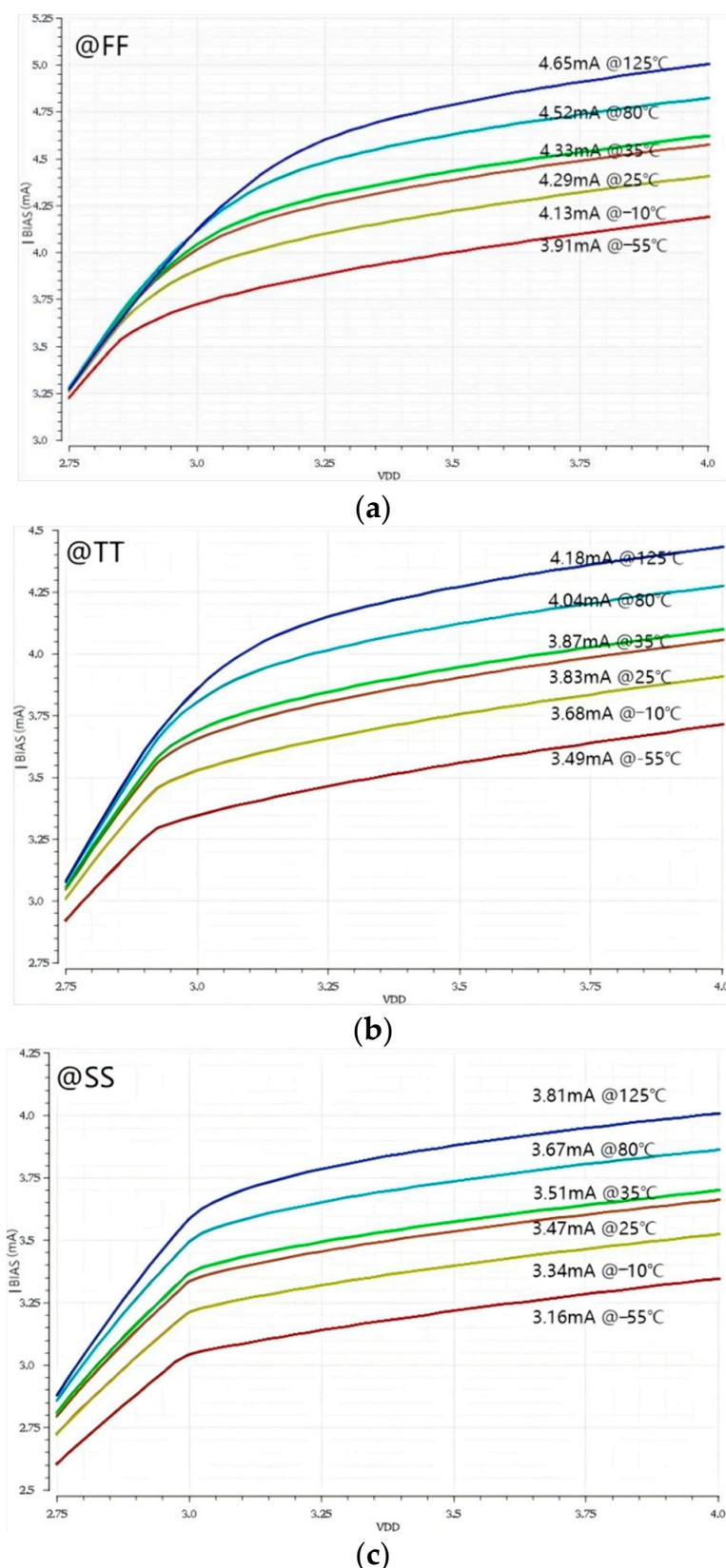

**Figure 15.** Simulated PVT variations of the bias current ($I_{BIAS}$) for (**a**) FF, (**b**) TT, and (**c**) SS corners, respectively.

Compared with a previous baseline design where the thermal variations in bias current exceeds 15.6%, the proposed bias circuit achieves significantly better thermal resilience. Across all the simulated PVT combinations, the current fluctuation remains confined within

$\pm10\%$, thus validating the effectiveness of the bandgap-stabilized architecture in enhancing current regulation under severe environmental stress.

Figure 16 displays the transient simulation results of the proposed bandgap-based CMVD across various PVT conditions. At the nominal corner (typical-typical, TT, at 27 °C), the peak output current reaches 14.67 mA$_{pp}$. Under the most favorable conditions (fast-fast, FF, at −55 °C), the maximum output current increases slightly to 15.39 mA$_{pp}$. Conversely, at the most stressful corner (slow-slow, SS, at 125 °C), the output current shows a minor drop to 14.25 mA$_{pp}$, indicating a worst-case degradation of only 7.4%.

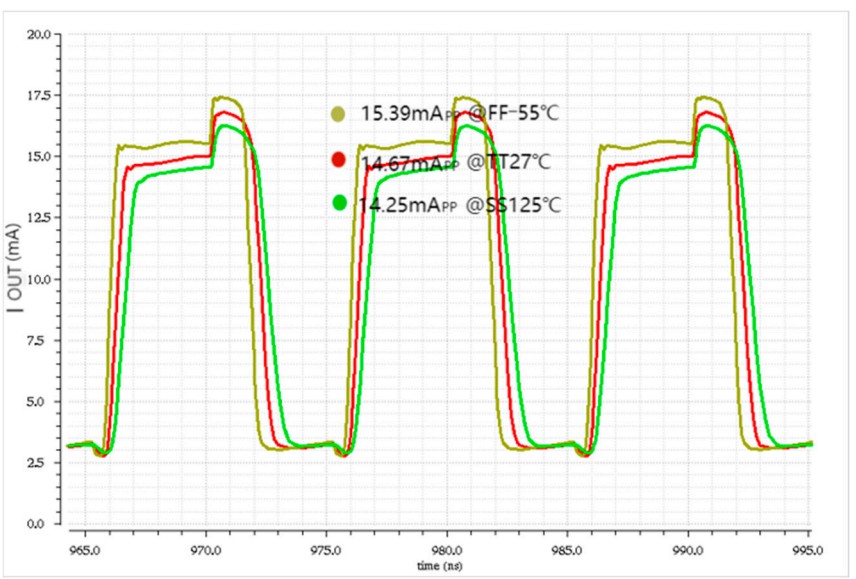

**Figure 16.** Transient output current waveform of the proposed CMVD under FF@−55 °C, TT@27 °C, and SS@125 °C conditions.

This performance confirms the robustness of the proposed CMVD under a wide range of temperature and process variations. Even in the worst-case scenario, the current swing remains within an acceptable margin, ensuring that sufficient modulation current is consistently delivered to the VCSEL diode for reliable optical emission.

Table 3 compares the key performance metrics of the proposed bandgap-based CMVD with several previously reported CMOS laser (or VCSEL) diode drivers [16–18], all of which were designed for low-power optical transmission using edge-emitting or VCSEL diodes. Although the proposed CMVD is implemented in a 180 nm CMOS process with a single 3.3 V supply, it demonstrates superior thermal stability and integration density. Notably, the worst-case PVT variations in its output current are significantly reduced to 9.8% compared to 23.5% in Ref. [17] and 14.3% in Ref. [16]. In addition, the proposed design achieves a lower power consumption (18 mW) than Ref. [16] (94 mW/channel) and comparable dissipation to Ref. [18] (12 mW), while maintaining a much smaller core area (0.06 mm$^2$).

In terms of the current-driving capability, the proposed CMVD supports up to 10 mA$_{pp}$ modulation currents with the average bias current of 4 mA, thus ensuring sufficient output strength for short-range LiDAR or sensing applications. While its maximum power consumption of 18 mW is slightly higher than the prior designs (11 mW), it reflects the enhanced output capability and stability under severe variations.

Moreover, the proposed CMVD demonstrates a compact layout footprint, occupying only 0.06 mm$^2$ in the core area, which is 40–70% smaller. This indicates excellent area efficiency, which is essential for future multi-channel optical transmitter integration in resource-constrained environments.

**Table 3.** Performance comparison with previously reported CMOS laser (or VCSEL) diode drivers.

| Parameters | [17] | [16] | [18] | This Work |
|---|---|---|---|---|
| CMOS technology (nm) | 350 | 65 | 65 | 180 |
| Supply (V) | 5 | 1.2/2.5/3.3 | 1.2 | 3.3 |
| Laser type | laser diode | VCSEL | VCSEL | VCSEL |
| Configuration | current-mode | voltage-mode | current-mode | current-mode |
| Output signaling | single-ended | single-ended | single-ended | single-ended |
| Driver type | common-cathode | common-cathode | common-cathode | common-cathode |
| Modulation current ($mA_{pp}$) | 50 (1 ch.) $\times$ 4 = 200 | 14 | 6.3 | 10 |
| Bias current (mA) | - | 2.1 | - | 4 |
| Power dissipation (mW) | - | 94/channel | 12 | 18 |
| Worst-case PVT variation | 23.5% | 14.3% | - | 9.8% |
| Core area ($mm^2$) | - | - | 0.4 | 0.06 |

Overall, the proposed CMVD shows competitive or superior performance across all key metrics, validating its suitability for compact, stable, and efficient VCSEL driving in short-range optical systems.

## 6. Conclusions

In this paper, a compact and temperature-stabilized current-mode VCSEL driver (CMVD) was implemented in a standard 180 nm CMOS process for short-range LiDAR applications in resource-constrained environments. By incorporating a bandgap-based biasing framework, the proposed CMVD significantly improves robustness against significant PVT variations while maintaining high modulation precision. Post-layout simulations confirm that the reference voltage shows excellent thermal stability, with its deviations confined within $\pm 2$%, and that the bias current exhibits less than 10% variations across the full temperature and process corners. Moreover, the transient output currents demonstrate strong resilience, with only a 7.4% drop under the worst-case SS@125 $^\circ$C condition. Compared to prior VCSEL driver implementations, the proposed CMVD achieves the lowest PVT variation (9.8%) and the most compact footprint (0.06 $mm^2$), while providing up to 10 $mA_{pp}$ modulation current and consuming only 18 mW. To further evaluate the CMVD's dynamic behavior, transient simulations were conducted using a 5 ns input pulse width. The measured rise and fall times of the output current under TT@27 $^\circ$C were approximately 1.52 ns and 0.62 ns, respectively, indicating fast switching capability and suitability for high-speed LiDAR modulation.

The circuit's high reliability, low power consumption, and small area characteristics make it an attractive solution for LiDAR-based sensing systems that demand privacy, cost efficiency, and thermal resilience—particularly in socially impactful use cases such as real-time behavior monitoring of children living in housing poverty. Future work will explore extending the architecture toward multi-channel arrays and integrating adaptive calibration for enhanced long-term stability and dynamic range.

**Author Contributions:** Conceptualization, S.-M.P.; methodology, S.-M.P. and J.L.; validation, J.L.; writing—original draft preparation, S.-M.P.; writing—review and editing, J.L. and S.-M.P.; visualization, J.L.; supervision, S.-M.P.; project administration, S.-M.P.; funding acquisition, S.-M.P. All authors have read and agreed to the published version of the manuscript.

**Funding:** This research was supported by the Ministry of Science and ICT (MSIT), Korea, under the Information Technology Research Center (ITRC) support program (IITP-2025-RS-2020-II201847) supervised by the Institute for Information and Communications Technology Planning and Evaluation (IITP).

**Institutional Review Board Statement:** Not applicable.

**Informed Consent Statement:** Not applicable.

**Data Availability Statement:** Data are contained within the article.

**Acknowledgments:** The EDA tool and chip fabrication were supported by the IC Design Education Center (IDEC), Korea.

**Conflicts of Interest:** The authors declare no conflicts of interest.

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
