# Peer review of "A CMOS Bandgap-Based VCSEL Driver for Temperature-Robust Optical Applications"

_photonics, doi:10.3390/photonics12090902_

Round 1

Reviewer 1 Report (Previous Reviewer 1)

Comments and Suggestions for Authors

The submitted article is a revised version of a previous submission, in which some adjustments have been made. The authors propose a driver for a VCSEL that is robust against temperature variations, aimed at short-range LIDAR system applications. In such applications, the fundamental parameter is the time-of-flight (TOF), which is used to calculate the distance to the object.

The main issue with the article lies in the fact that the authors present only one block of the proposed application (the driver), without demonstrating how it directly contributes to improving TOF accuracy. Since TOF is the key parameter, the authors must justify how the use of a band-gap reference enhances measurement precision. Otherwise, the LIDAR application becomes irrelevant, and the focus should be solely on the band-gap reference and its impact on current accuracy and modulation linearity.

In addition to this, several points require improvement or clarification, including:

  • The quality of all figures must be improved, especially the Y-label in all the graphs.
  • Lines 68–71: The mentioned statement does not correspond to Figure 2. A revision of the text is needed to ensure consistency.
  • Figure 2: What is the typical current required by a VCSEL? For what current was the proposed driver designed?
  • Lines 118–120: If TOF is the critical parameter, why is the accuracy of the current mirror emphasized? A detailed explanation is needed to relate this point to TOF measurement.
  • Figures 3 and 4: A quantitative explanation (with equations) is required to show the difference in TOF accuracy between the conventional and proposed drivers.
  • What elements affect TOF accuracy in the conventional configuration (Figure 3) and in the proposed one (Figure 4)? Light intensity does not alter time-of-flight, so what is the actual improvement?
  • Figure 4:
    • According to lines 146 and 164–165, the band-gap block generates VBIAS and VREF. However, in Figure 5, the block receives VBIAS to generate VREF. This suggests an error in Figure 5.
    • The signal “S” is not described upon its first appearance in the figure.
  • Figure 5:
    • A complete temperature characterization is required.
    • The table indicates that the circuit is impractical and highly sensitive to mismatch. The dimensions do not allow layout techniques such as interdigitated or common centroid.
    • Does “Active region” refer to the linear/ohmic region?
    • The OPAMP output appears to be disconnected.
    • Separate the table from the schematic and label it as Table I.
  • Figure 6: Separate the table from the schematic and label it as Table II.
  • Figure 7:
    • The IB block appears to be an AND gate. Why is it presented as a buffer? What additional function does it serve?
    • Lines 155–157: It is recommended to repeat the measurements using a real VCSEL diode on the PCB.
    • Line 167: Add references regarding the Brokaw cell.
  • Figure 8: The IB block does not appear in the layout. Why?
  • Line 250: It is mentioned that a chip has been fabricated. Why are no experimental results presented instead of simulations only? The authors should present measurement results.
  • Equation (2): What does “m” represent? Include its mathematical expression.
  • Figure 9: Does the Y-axis represent VREF or VBIAS?
  • Figure 10:
    • The graph indicates a unity-gain bandwidth close to 100 MHz, not 33 MHz. Review the graph and simulation.
    • The compensation appears incorrect: the system goes through a zero-pole-pole-zero sequence before the unity-gain point.
  • Lines 267–268: Verify that the data matches the graph.
  • Figure 11:
    • Does the Y-axis represent VBIAS or VREF?
    • The 23.8% value suggests poor performance of the band-gap circuit. Express the variation in ppm and compare it with other works in Table I.
  • Lines 276–277: It is unclear whether there is effective first- and second-order compensation. Expand the explanation.
  • Figure 12: Label the Y-axis correctly.
  • Figure 13: Label the Y-axis (VREF or VBIAS?) and each curve.
  • Figure 14:
    • Label the Y-axis correctly (VREF).
    • Lines 300–309: The graphs show poor circuit behavior, with values below 1.2 V when the nominal is 1.71 V.
    • Verify the percentages reported in lines 300–309 and 310–312, as they do not match the graphs.
  • Figure 15: Similar observations apply as in Figure 14.
  • Table I: In reference [19], the author compares against their own work. This must be corrected.

Typos:

  • Line 28: “traction” → should it be “attraction”?

Comments on the Quality of English Language

No comments.

Author Response

Dear Reviewer 1,

Please find the attached file that is an answer sheet for your valuable comments.

Best regards,

Sung Min Park

Reviewer 2 Report (New Reviewer)

Comments and Suggestions for Authors

Overall this manuscript is well-written and easy to read, and the design considerations seem reasonable enough, but the presented circuits would appear to have limited novelty.  Bandgap references are very commonly used to extend the thermal range for analog circuits, so their use in this circuit is not surprising.  The op-amps are textbook two-stage topologies.  Also, the comparisons in Table 1 are quite meager; for more data, refer to Mowlavi: "A Review of IC Drivers for VCSELs in Datacom Applications", IEEE Transactions on VLSI Systems, 2023.  

The LIDAR safety application is new to me.  I would have expected to see some discussion of the speed requirements for the driver -- I assume these would be derived from system specifications. Speed requirements often determine the viable circuit solutions, regardless of the application.  

The simulated temperature sweeps cover a range much larger than what the application would seem to need.  Also, a truly temperature-insensitive solution would need to take into account also the severe temperature dependence of the VCSEL.  

Page 2, line 68: this is not the content of the figure referred to. 

Fig 3: I would expect a reference for this driver design. 

Page 5: in the figure, the op-amp output does not seem to be connected to anything?  If its output connection is that shown on page 6, why are the CMOS device labels not identical? 

Author Response

1. Overall, this manuscript is well-written and easy to read, and the design considerations seem reasonable enough, but the presented circuits would appear to have limited novelty. Bandgap references are very commonly used to extend the thermal range for analog circuits, so their use in this circuit is not surprising. The op-amps are textbook two-stage topologies. Also, the comparisons in Table 1 are quite meager; for more data, refer to Mowlavi: "A Review of IC Drivers for VCSELs in Datacom Applications", IEEE Transactions on VLSI Systems, 2023.

--> (ans.) We appreciate the reviewer’s thoughtful feedback. It is true that bandgap references and two-stage op-amps are established techniques in analog IC designs. However, the novelty of this work lies in the integration of a bandgap-stabilized bias within a VCSEL driver architecture, thereby enabling the improved thermal robustness, modularity, and low power/area implementation—attributes that are not commonly achieved together in the literature. Also, we have revised Table I (Table III in the revised manuscript) by incorporating additional comparative data drawn from the relevant review article (in [19]). Mowlavi et al., “A Review of IC Drivers for VCSELs in Datacom Applications,” IEEE Transactions on VLSI Systems, 2023.

2. The LiDAR safety application is new to me. I would have expected to see some discussion of the speed requirements for the driver -- I assume these would be derived from system specifications. Speed requirements often determine the viable circuit solutions, regardless of the application.

--> (ans.) We fully agree that speed requirements are typically derived from system-level specifications and often play a decisive role in guiding the choice of circuit topology. Yet, in our paper, the primary objective was to address the challenges of thermal stability, bias robustness, and compact implementation for VCSEL drivers in sensing-oriented applications. For this reason, the analysis and discussion mainly emphasize the temperature stability and process insensitivity rather than speed optimization.

Nevertheless, we acknowledge that incorporating a discussion on the relationship between system-level speed requirements and driver design choices would enrich the completeness of the work. While such an analysis was beyond the immediate scope of the current manuscript, we very much appreciate the reviewer’s suggestion and will consider it carefully in our future research and publications, where a more thorough treatment of speed constraints and their influence on the driver architecture will be provided.

3. The simulated temperature sweeps cover a range much larger than what the application would seem to need. Also, a truly temperature-insensitive solution would need to take into account also the severe temperature dependence of the VCSEL.

--> (ans.) We appreciate the reviewer’s valuable observation. It is true that the simulated temperature sweeps cover a range wider than what may be strictly required for the intended sensing application. Our intention in doing so was to demonstrate the robustness of the proposed driver under the extreme operating conditions, thereby ensuring reliability even beyond the nominal system requirements.

We also agree with the reviewer that a truly temperature-insensitive solution should consider not only the biasing circuitry, but also the inherent thermal dependence of the VCSEL device itself. In the present work, our focus has been limited to improving the thermal robustness of the driver circuit, while the VCSEL characteristics were assumed to be ideal for the purpose of circuit-level evaluation. We acknowledge that including the VCSEL temperature dependence would provide a more complete system-level picture, and we consider this an important direction for our future research.

4. Fig 3: I would expect a reference for this driver design.

--> (ans.) The conventional current-mode VCSEL driver (shown in Figure 3) was originally included with a corresponding reference, but the citation was mistakenly omitted in the previous version of the manuscript. This has now been corrected, and the proper reference has been added in the revised manuscript (Ref. [8], Line 139).

5. Page 5: in the figure, the op-amp output does not seem to be connected to anything? If its output connection is that shown on page 6, why are the CMOS device labels not identical?

--> (ans.) The op-amp output in Figure 5 is now connected to the bias node (VBIAS) in the revised schematic.

Regarding the reviewer’s second comment, the CMOS device labels between Figure 5 and Figure 6 are not identical because these two figures are different circuit blocks. Figure 5 is the bandgap reference core, whereas Figure 6 is the driver stage. While the op-amp topologies may appear similar, they were designed and labeled independently.

Round 2

Reviewer 1 Report (Previous Reviewer 1)

Comments and Suggestions for Authors

The manuscript has been improved, but there are still several things that are wrong, for example:

- Reference 14 is incorrectly written.
- References [14] through [16] are unrelated to the Brokaw cell.

This manuscript is a resubmission of an earlier submission. The following is a list of the peer review reports and author responses from that submission.

Round 1

Reviewer 1 Report

Comments and Suggestions for Authors

The core of the work focuses on a stabilized band-gap reference used in a power-optimized VCSEL driver. Therefore, it is suggested to rewrite the article by focusing on this block, analyzing it in detail, and providing the corresponding mathematical justification. The specific application (LIDAR in this case) should be removed, as the proposed solution can be applied in multiple scenarios.

Other issues follows:

  • There is no significant contribution in the work.
  • The proposed circuit lacks the merits required for publication.
  • According to the comparison table, the results are mainly compared against two other publications by the same author(s).
  • The improvement claimed in the work is limited to PVT variation, which is neither detailed nor mathematically justified; only post-layout simulations are presented.
  • Out of the 19 references, 6 are authored by one of the contributors. Others are outdated, such as [8] and [16].

Additionally,

The references include several citations to one or more of the authors, for example, references [3], [6], [7], [10], [18], and [19].
Table [1]: The authors compare against their own work in 2 out of 3 citations.
Regarding the PDK, the foundry and the process flavor must be specified.

The first time the VCSEL was used, it was not defined.

Line 40: The statement “The paper proposes a compact and thermally robust LiDAR transmitter” is inaccurate because the paper focuses solely on the driver. Please rewrite lines 40–44 accordingly.

Line 44: The authors claim that the proposed driver is tolerant to PVT (Process, Voltage, Temperature) variations; however, the paper does not describe any specific techniques used to achieve this—only feedback is mentioned. Please include a subsection that provides a detailed explanation supporting this claim.

Figure 1: In this specific application, the laser light points in a single direction. How is the issue addressed of a child moving around a room and the laser tracking them? How is the problem solved for the VCSEL diode to perform tracking on the subject to be monitored?

Figure 1: LDD is not defined.

Lines 120–122: If the circuit is robust to PVT variations due to the use of a feedback loop, then an analysis of the feedback loop should be presented, including its gain and stability, and how these parameters improve the circuit’s performance under PVT variations.

Line 170: This statement is incorrect; the band-gap reference cannot be temperature-insensitive. At best, it reduces temperature dependence.

Figure 5: The circuit is not novel, so there is no contribution.
Figure 5: For transistors T1 and T2, the terminal B (base) is floating.
Figure 5: The transistor dimensions and the operating point must be included. Also, the design considerations that justify the transistor sizing must be included.
Line 210: The analysis and the expressions for IREF and VREF must be included. Also, the finite gain of the OPAMP must be considered in the expressions for I_REF and V_REF.

Figure 6: The transistor dimensions and the operating point must be included!
Figure 6: Include a table with the OPAMP characteristics. What is the value of C?
Line 242: The foundry and the process flavor must be specified.

Line 243: What was the setup used to calculate the power consumption, and what type of power is being referred to (mean, RMS, etc.)?

Line 251: What types of transistors are being used? Typically, 180nm technology only provides 1.8V and 3.3V transistors.

Line 260: Consider using “stability” or “insensitivity” for clarity.

Figure 9(a): The total error for the curve at 125°C is approximately (1.191 – 1.123)/1.17 = 5.8%.

In addition to Figure 9, temperature sweep plots (temperature on the X-axis vs. V_REF on the Y-axis) as a function of VDD should be included. The same applies to Figure 10 (temperature sweep on the X-axis vs. IBIAS on the Y-axis).

Figure 11: The Y-axis is incomplete, and there are no labels.

Line 272: It seems that no specific techniques were applied to make the circuit robust to PVT variations. Aside from feedback, what other methods were used to ensure PVT robustness?

Table I: The proposed circuit exhibits the highest power consumption. The only improvement is with respect to PVT variation. Regarding area, it is difficult to assess since no transistor or capacitor sizes are provided.

Reviewer 2 Report

Comments and Suggestions for Authors

The manuscript presents a well-­engineered bandgap-stabilized, current­-mode VCSEL driver (CMVD) that demonstrates excellent thermal stability and compact area in a standard 180-nm CMOS process, and the overall structure and simulation results are convincing. However, the paper would benefit from a deeper treatment in several areas to strengthen both its technical impact and readability.

First, the Introduction could be refined to focus more on the specific challenges of PVT variation in current-mode drivers, briefly summarizing existing bandgap-reference approaches, and then clearly stating the novelty of integrating digital current-steering logic with a Brokaw-cell bias in a single, compact core. This could help the reader grasp the unique contribution of the work without extensive background on LiDAR applications.

In terms of depth, the paper relies entirely on post-layout simulations. Presenting preliminary silicon measurement data, such as even a small set of room-temperature and elevated-temperature measurements, would greatly enhance confidence in the real-world performance of the proposed design. If wafer results are not yet available, consider improving the simulation section with corner-based Monte Carlo analyses to quantify sensitivity to device mismatch and to show how process skew and random variations affect the key metrics (VREF drift, IBIAS deviation, modulation current accuracy).

The clarity of presentation can also be improved by consolidating the descriptions of each subcircuit into a single, streamlined narrative rather than repeating similar block­level explanations. For instance, the discussion of the Brokaw bandgap and startup circuitry in Section 4 could be shortened and cross-referenced to well-known literature, allowing more space to elaborate on the bias-generator’s op-amp compensation scheme (as shown in Figure 6) and its stability margins.

Similarly, Figures 9 to 11 would benefit from unified axis labels, inclusion of numerical legends directly on the plots, and error bars or shaded regions indicating expected variation across process corners.

In the Results section, a brief comparison of rise/fall time trade-offs, linearity (e.g., THD or modulation amplitude vs. frequency), or jitter performance would provide a fuller picture of the driver’s dynamic behavior, which is critical for LiDAR applications.

Finally, there are too many self-citations in the reference list. In addition, it is not up to date.

Comments on the Quality of English Language

A professional copy edit would resolve occasional grammatical inconsistencies and improve readability. For instance, replacing passive constructions like “this work proposes” with more direct phrasing such as “the proposed CMVD integrates.” Consistent tense usage and careful definition of acronyms at first use will help non­expert readers follow the argument. 

Reviewer 3 Report

Comments and Suggestions for Authors

The introduction section presents a potential application of lidar. However, the connection between this application and the current article is not clearly explained.

“This becomes particularly problematic in indoor environments with poor thermal regulation, such as those found in housing-poverty scenarios. ” In terms of the requirements for the drive circuit, it is hard to say that this environment brings higher demands. Whether it is for mobile phones or automotive in-vehicle applications, the environmental conditions required are more stringent than those of this environment.

Basically, I think the author has not found a suitable foothold to carry out the work. Some of the problems mentioned in the text do exist, but they can be easily solved through other means.

The circuit proposed by the author, according to the simulation results, can enhance stability. However, its power consumption has increased significantly compared to those from the references. Moreover, among the three objects for performance comparison, two are the author's previous articles, and the other one is the driver of edge-emitting lasers in 2011, which does not represent the current mainstream performance of the drivers.